# FROM SLOTS TO MASKS: RETHINKING OCL

## ABSTRACT

Object-centric learning (OCL) aims to learn unsupervised representations that isolate individual objects from their context, motivated by goals such as out-of-distribution (OOD) generalization, compositional generalization, and structured environment modeling. Most prior work has developed slot-based mechanisms for object separation, typically evaluated on unsupervised object discovery.

Recent advances in segmentation provide a scalable alternative: class-agnostic models can separate objects directly in pixel space, enabling independent encoding. We show that such segmentation-based approaches achieve strong zero-shot performance on OOD object discovery benchmarks, scale naturally to foundation models, and flexibly handle a variable number of objects. For the task of object discovery, segmentation therefore offers a practical substitute for slot-based OCL. A broader question is how object separation contributes to downstream goals. We address this in the setting of OOD robustness, focusing on spurious background correlations. We introduce a training-free probe, **Object-Centric Classification with Applied Masks** (**OCCAM**), and find that segmentation-based encodings of individual objects improve robustness compared to slot-based OCL methods. Our study does not address compositional generalization or reasoning tasks directly, but provides a complementary benchmark where object-centric representations deliver tangible benefits. We release our code and tools to enable the community to explore segmentation-based object-centric representations at scale, and to support practical applications of OCL beyond object discovery.

## 1 INTRODUCTION

Object-centric learning (OCL) seeks to develop representations of complex scenes that independently encode each foreground object separately from background cues, ensuring that one object's representation is not influenced by others or the background (Greff et al., 2019; Burgess et al., 2019). This constitutes a foundational element for many objectives: it supports modeling of structured environments (Schölkopf et al., 2021), enables robust out-of-distribution (OOD) generalization (Dittadi et al., 2022; Arefin et al., 2024; Wiedemer et al., 2024; Mamaghan et al., 2025; Kapl et al., 2025), facilitates compositional perception of complex scenes (Greff et al., 2020), and deepens our understanding of object perception in human cognition (Spelke, 1990a; Téglás et al., 2011; Wagemans, 2015). However, despite these broad goals, most research in OCL has centered on advancing "slot-centric" methods that separate objects and encode them into slots, evaluated using unsupervised object discovery as the primary metric (Locatello et al., 2020; Jiang et al., 2023; Seitzer et al., 2023; Didolkar et al., 2025; Kipf et al., 2022; Elsayed et al., 2022; Greff et al., 2019). In this paper, we challenge the continued emphasis on developing mechanisms to separate objects in representation space as the main challenge to be addressed in OCL.

We first show that sample-efficient class-agnostic segmentation models, such as High-Quality Entity Segmentation (HQES) (Lu et al., 2023) are far stronger than the latest slot-centric OCL approaches, already achieving impressive zero-shot object discovery. Moreover, these models are scalable, with foundation models like Segment Anything (SAM) (Kirillov et al., 2023; Ravi et al., 2025) showing remarkable zero-shot segmentation, addressing much of what is usually tackled with slot-centric approaches. While prior work has occasionally used predicted masks in OCL pipelines, segmentation has not been systematically proposed as a substitute for slot-based discovery. This perspective motivates a broader question: How does the ability to separate objects within scenes contribute to other OCL objectives, such as OOD generalization?

We bridge this gap by directly linking OCL to OOD generalization, especially in known hard settings with spurious background cues. We introduce **Object-Centric Classification with Applied Masks** (**OCCAM**), a simple, object-centric probe for robust zero-shot image classification. OCCAM consists of two stages: (1) generating object-centric representations via object-wise mask generation, and (2) applying OCL representations to downstream applications, such as image classification in the presence of spurious backgrounds, by selectively focusing on relevant object features while discarding misleading background cues.

Empirically, we find that, on Stage (1), sample-efficient segmentation models outperform current OCL approaches in obtaining object-centric representations without additional training. However, on Stage (2) — the task of identifying relevant object cues amidst numerous possible masks — remains a challenge. Nevertheless, when Stage (2) is executed correctly, simple OCL probes such as OCCAM already have the potential for robust OOD generalization.

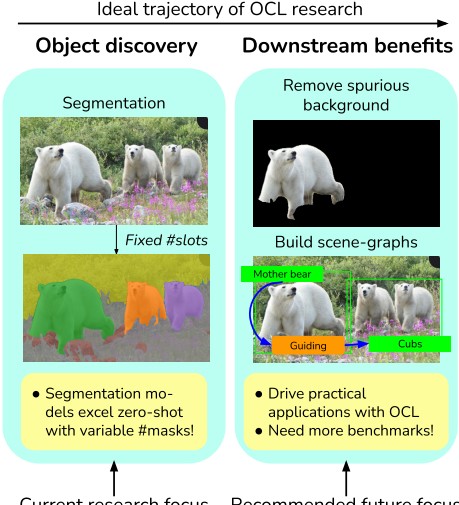

Figure 1: **Where Should We Go?** OCL progress is mostly measured on benchmarks where foundational segmentation models already excel in a zero-shot setting, eliminating the need for unsupervised methods. We call for new benchmarks assessing the downstream efficacy of OCL and consider robustness to spurious backgrounds as one example.

We recommend more focus by future OCL works on creating benchmarks, methodologies for testing real-world applications where object-centric representations offer clear practical benefits, encouraging theory motivated by specific real-world tasks.

## 2 RELATED WORK

We review object-centric learning (OCL) from three angles: motivation, evaluation, and methodologies. For extended related work see § A.

**Motivation.** OCL is motivated by several perspectives: uncovering latent generative factors (e.g., position, color) (Fumero et al., 2023), capturing causal structure (Liu et al., 2023b; Schölkopf et al., 2021), modeling human cognition (Spelke, 1990a; Téglás et al., 2011; Wagemans, 2015), and even mimicking how infants learn to track objects (Dittadi et al., 2022). It also aims to model scene compositionality (Greff et al., 2020), with claims of improved efficiency, generalization, and robustness (Locatello et al., 2020; Kipf et al., 2022; Seitzer et al., 2023; Wiedemer et al., 2024; Mamaghan et al., 2025; Kapl et al., 2025; Arefin et al., 2024). Yet the empirical evidence that OCL really achieves these goals or brings substantial downstream benefits is very limited. We address this gap by demonstrating OCL benefits for robust classification.

**Evaluation.** Despite claims of efficiency and generalization (Kipf et al., 2022; Kapl et al., 2025; Arefin et al., 2024; Wiedemer et al., 2024), OCL lacks scalable benchmarks. The progress is mostly tracked via unsupervised object discovery (Locatello et al., 2020; Jiang et al., 2023; Seitzer et al., 2023; Didolkar et al., 2025; Kipf et al., 2022; Elsayed et al., 2022; Greff et al., 2019). We argue for broader evaluations, as modern foundational segmentation models now surpass OCL in discovery (see Table 1, Figure 3).

**Methodologies.** SlotAttention (Locatello et al., 2020) popularized iterative slot-based representations. Its variants include Dinosaur (Seitzer et al., 2023; Didolkar et al., 2025), which leverages DINO features (Caron et al., 2021; Dosovitskiy et al., 2021), and SlotDiffusion (Jiang et al., 2023), that combines slots with diffusion decoders (Rombach et al., 2022). OCL has also been studied for OOD segmentation (Dittadi et al., 2022), compositional generalization (Mamaghan et al., 2025; Kapl et al., 2025), and robust classification via CoBalT (Arefin et al., 2024). In our experiments, we compare against SlotDiffusion, (FT-)Dinosaur, and CoBalT.

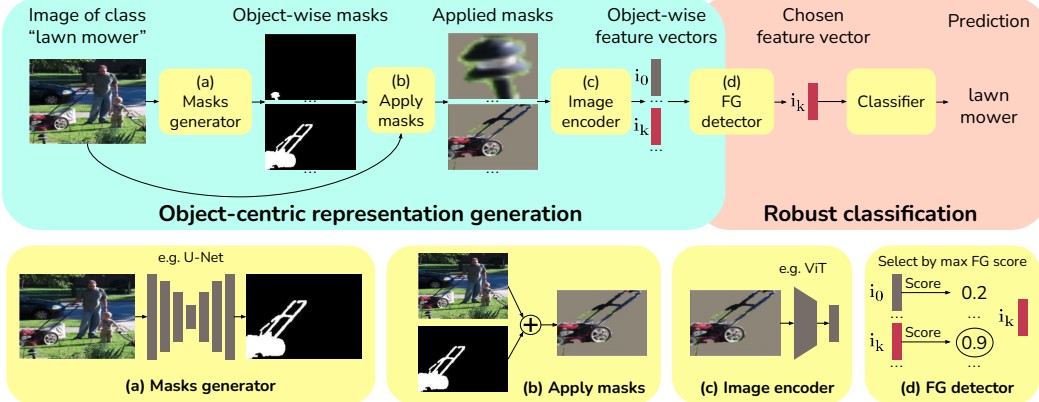

Figure 2: **Overview of Object-Centric Classification with Applied Masks (OCCAM)**. There are two main parts. The first part (§ 3.1) uses entity segmentation masks for **object-centric representation generation**. The second part (§ 3.2) performs **robust classification** by selecting representations corresponding to the foreground object and using them for classification. Indices $[i_0, \ldots, i_k, \ldots]$ correspond to each object in the scene.

## 3 METHOD

**Notations.** We denote an image as $x \in [0, 1]^{[3, H, W]}$ and a label as $y \in \mathcal{Y} = \{1, \ldots, C\}$, where $C$ is the number of classes. We will write an image encoder, or a feature extractor, as $\psi$ and image embedding, or feature vector, as $\psi(x) \in \mathbb{R}^d$, where $d \geq 1$ is the feature dimensionality. We define the classifier's pre-softmax logits as $f(\psi(x)) \in \mathbb{R}^{|\mathcal{Y}|}$ and softmax probabilities as $p(\psi(x)) = \mathrm{Softmax}(f(\psi(x))) \in [0, 1]^{|\mathcal{Y}|}$. For simplicity, we will use $p(\psi(x))$ and $p(x)$ interchangeably. We also denote indices for the last two dimensions in tensors as superscripts (e.g., last two dimensions of sizes H, W for $x$) and all other dimensions as subscripts (e.g., first dimension of size 3 in $x$). We will use shorthands "FG" and "BG" for foreground and background, respectively.

Our Object-Centric Classification with Applied Masks (OCCAM) pipeline is summarized in Figure 2. We use object-centric representations to reduce spurious correlations in image classification. It consists of two main parts: 1. generate object-centric representations, 2. perform robust classification by classifying an image using only representations of the foreground object. In the following subsections, we will explain these parts in more detail.

### 3.1 GENERATING OBJECT-CENTRIC REPRESENTATIONS

To generate the object-centric representations, we first generate masks for all objects and backgrounds in the image using a mask generator. We then apply generated masks to images by combining masks with images. Each object is then encoded with an image encoder.

**Generating masks.** To produce object representations given an original image $x \in [0, 1]^{[3, H, W]}$, we generate a set of masks for all the foreground objects and the background. That is done with the help of a mask generator $S$, which takes $x$ as input and assigns each pixel in $x$ to one of $K_{\max}$ masks. The output of this model is the stack of $K$ binary masks, with each mask $m$ corresponding to a different object: $m \in \{S_i, i = 1 \ldots K\}, m \in \{0, 1\}^{[H, W]}$. An OCL method like FT-Dinosaur (Didolkar et al., 2025) or an external segmentation model like High-Quality Entity Segmentation (HQES) (Lu et al., 2023) can be used as a mask generator in this pipeline. We will call the mask generator as the mask model or the masking method interchangeably.

**Applying masks.** After producing the binary masks for each object, we segregate the pixel contents for each mask by applying the mask on the input image. We will interchangeably call the mask applying operation as the mask method throughout the paper. One way to apply masks to images is to simply add a gray background to all but selected pixels, cropping the image that follows the mask contours, and resizing the result to the size of the original image. In such a case, we call the operation "Gray BG + Crop".

However, a mask method can be any operation involving an image $x$ and a mask $m$: $a(x, m) \in [0, 1]^{[3, H, W]}$. We additionally show ease-of-use in incorporating the latest masking techniques like AlphaCLIP, which combines a mask and original image by appending masks as an additional $\alpha$-channel to the image tensor, resulting in an RGB-A 4-dimensional tensor. This allows using masks as a source of focus instead of removing backgrounds entirely, useful for some practical applications. We call such an operation as "$\alpha$-channel".

**Encoding applied masks.** To get the final object-centric representations we encode applied masks by an image encoder $\psi$ such as ViT (Dosovitskiy et al., 2021) for example.

### 3.2 ROBUST CLASSIFIER

We hypothesize that by isolating foreground object representations from the representations of background and other objects, we eliminate sources of spurious correlations, hence performing more robust classification. For that reason, we first use the set of object-centric representations obtained in the previous stage to select the single representation that corresponds to the foreground. Then we provide the selected foreground representation to the classifier to make the final prediction.

**FG detector.** After applying masks to the image, we select the mask that corresponds to the foreground object by the following process. At first, we compute the *foreground score* that reflects how likely a given applied mask is to correspond to the foreground object. Then we take the mask with the highest foreground score among all masks for the current image and use it for robust classification.

Currently, we use two types of foreground scores, both computed from the classifier's outputs:

1. **Ens.** $\mathcal{H}$: $g_{\mathcal{H}}(x, m) = \frac{1}{M} \sum_{k=1}^{M} \mathcal{H}[p_k(\psi(a(x, m)))]$ - ensemble entropy (see details in § B). Here, $M$ is the ensemble size, and $\mathcal{H}$ stands for entropy.
2. **Class-Aided**: $g_{\text{class\_aided}}(x, m) = p^y(\psi(a(x, m)))$ - probability of predicting a ground truth label. We consider this foreground score to measure the efficacy of the object-centric representation rather than to suggest it as a final method to use in practice. Although in reality, we do not have access to ground truth labels, it provides critical signals as to whether the insufficient generalization performance is due to object representation or due to foreground selection and the classifier.

For the comparison of different foreground scores, see § B.

**Image classification using FG object representations.** Finally, once we have identified the mask that matches the foreground object, we apply it to the original image and classify the result of this operation. The final output of our method is: $\text{OCCAM}(x) = p(\psi(a(x, m^\star))$, where $m^\star$ is the mask selected by the FG detector.

## 4 EXPERIMENTS

In this section, we first evaluate slot-centric OCL approaches and foundational segmentation models on unsupervised object discovery tasks. We then evaluate whether OCL methods provide robust object classification by benchmarking them against a strong baseline that uses mask predictions from foundational segmentation models, following the OCCAM pipeline (§3).

### 4.1 ARE WE DONE WITH OBJECT-DISCOVERY?

OCL methods are often evaluated by how well they perform on unsupervised object discovery, measured via instance segmentation for every object in the scene. We explore whether the emergence of strong zero-shot segmentation models (class-agnostic) such as HQES (Lu et al., 2023) and SAM (Kirillov et al., 2023) allows reliable decomposition of the scene into objects. We compare these foundational segmenters against state-of-the-art OCL approaches (Jiang et al., 2023; Seitzer et al., 2023; Didolkar et al., 2025).

**Setup.** We first describe our experimental setup, including datasets, metrics, and compared baselines. Following prior work (Kipf et al., 2022; Elsayed et al., 2022; Seitzer et al., 2023; Didolkar

| Image | DINOSAUR | SlotDiffusion | FT-DINOSAUR | SAM | HQES |
|---|---|---|---|---|---|

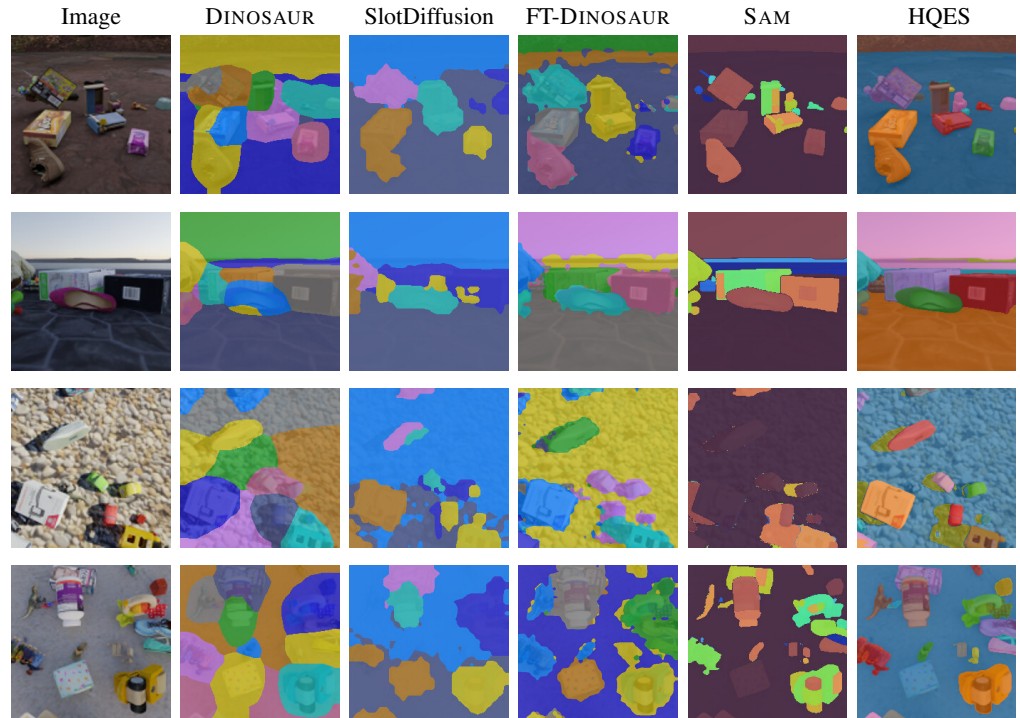

Figure 3: **Qualitative Results on Object Discovery**. DINOSAUR, SlotDiffusion, and FT-DINOSAUR are existing object-centric learning (OCL) approaches. SAM and HQES refer to zero-shot segmentation methods. Images are from MOVi-E. SAM and HQES masks fit objects much better than the masks predicted by OCL methods. All columns except for HQES are taken from (Didolkar et al., 2025).

et al., 2025), we use two synthetic image datasets from (Greff et al., 2022): Movi-C and Movi-E. Both feature around 1,000 realistic 3D-scanned objects placed on high-definition backgrounds. Movi-C contains 3 – 10 objects per scene, while Movi-E contains 11 – 23. We quantify model performance using two standard metrics (Table 1): the foreground adjusted Rand index (FG-ARI) (Rand, 1971; Hubert & Arabie, 1985; Kipf et al., 2022) and mean best overlap (mBO) (Pont-Tuset et al., 2015; Seitzer et al., 2023), detailed in Section §2. Unlike FG-ARI, mBO also accounts for background pixels. It also measures how well masks fit objects. We compare HQES and SAM to state-of-the-art OCL methods with demonstrated real-world applicability: SlotDiffusion (Jiang et al., 2023), Dinosaur (Seitzer et al., 2023), and FT-Dinosaur (Didolkar et al., 2025), all described in Section §2.

**Results.** Table 1 and Figure 3 show quantitative and qualitative results. Across both metrics, FG-ARI and mBO across out-of-distribution benchmarks like Movi-C and Movi-E, HQES far surpasses the OCL baselines. This gap is especially notable in mBO on Movi-E, improving 29.9% to 63.8%. Qualitatively, HQES masks fit objects much better than masks predicted by OCL methods (Figure 3). HQES also shows it is possible to be sample efficient, only being trained on 151k samples in contrast to 11M samples for SAM.

**Conclusion.** Sample-efficient segmentation models, even in a zero-shot setting, excel at object discovery, surpassing OCL methods by large margins. This suggests that one key aspect of OCL — decomposing the scene into objects — be largely solved by powerful pre-trained segmentation models, effectively replacing the slot-based OCL methods. Given the decomposition, we explore in the next section downstream applications where OCL methods can contribute a lot of practical value.

### 4.2 APPLICATION: CLASSIFICATION WITH SPURIOUS BACKGROUND CORRELATIONS

As foundational segmentation models outperform OCL methods in decomposing the scene into constituent objects, we take a further step and evaluate OCL methods on a downstream task that

| Method | Pre-train | | FT | Movi-C | | Movi-E | |
| --- | --- | --- | --- | --- | --- | --- | --- |
| | Encoder | Decoder | | ARI | mBO | ARI | mBO |
| Slot Diffusion (Jiang et al., 2023) | OpenImg (1.9M) | COCO (118k) | ✗ | 66.9 | 43.6 | 67.6 | 26.4 |
| Dinosaur (Seitzer et al., 2023) | GLD (1.2M) | COCO (118k) | ✗ | 67.0 | 34.5 | 71.1 | 24.2 |
| FT-Dinosaur (Didolkar et al., 2025) | GLD (1.2M) | COCO (118k) | ✓ | 73.3 | 44.2 | 71.1 | 29.9 |
| HQES (Lu et al., 2023) (Ours) | COCO (118k) + EntitySeg (33k) | | ✗ | 79.3 | 65.4 | **87.2** | 63.8 |
| SAM (Kirillov et al., 2023) | SA-1b (11M) | | ✗ | **79.7** | **73.5** | 84.7 | **69.7** |

Table 1: **Object Discovery Performance.** Quantitative results for object discovery on Movi-C and Movi-E; column "FT" indicates whether the model was fine-tuned on the training split of the corresponding dataset (Movi-C or Movi-E). HQES outperforms the OCL baselines like Slot Diffusion and Dinosaur, despite being sample-efficient (151k training samples).

**(a) ImgNet-D (BG)**

| Method | Acc. (↑) |
| --- | --- |
| CLIP ViT-L | |
| CLIP | 23.5 |
| O-D (Ours) | 57.7 |
| O-H (Ours) | 68.0 |
| CLIP-SigLip | 59.4 |
| O-D-SigLip (Ours) | 71.5 |
| O-H-SigLip (Ours) | **78.5** |
| Multi-modal LLMs | |
| MiniGPT-4 | 71.8 |
| LLaVa | 52.9 |
| LLaVa-NeXT | 68.8 |
| LLaVa-1.5 | 73.3* |

**(b) UrbanCars**

| Method | WGA (↑) |
| --- | --- |
| ViT-L-14 CLIP | |
| CLIP | 87.2 |
| O-D (Ours) | 98.4 |
| O-H (Ours) | **100.0** |
| ResNet50 CLIP | |
| CLIP | 64.8 |
| O-D (Ours) | 98.4 |
| O-H (Ours) | **100.0** |
| ResNet50 | |
| CoBalT | 80.0 |
| LfF | 34.0 |
| JTT | 55.8 |
| SPARE | 76.9 |
| LLE | 90.8* |

**(c) ImgNet-9 (MR)**

| Method | Acc. (↑) |
| --- | --- |
| ViT-L-14 CLIP | |
| CLIP | 91.9 |
| O-D (Ours) | 93.8 |
| O-H (Ours) | **95.2** |
| ResNet50 CLIP | |
| CLIP | 81.1 |
| O-D (Ours) | 80.6 |
| O-H (Ours) | **85.6** |
| ResNet50 | |
| CoBalT | 80.3 |
| SIN | 63.7 |
| INSIN | 78.5 |
| INCGN | 80.1 |
| MaskTune | 78.6 |
| CIM | 81.1* |

**(d) Waterbirds**

| Method | WGA (↑) |
| --- | --- |
| ViT-L-14 CLIP | |
| CLIP | 83.6 |
| O-D (Ours) | 92.1 |
| O-H (Ours) | **96.0** |
| ResNet50 CLIP | |
| CLIP | 72.9 |
| O-D (Ours) | 83.3 |
| O-H (Ours) | **92.5** |
| ResNet50 | |
| CoBalT | 90.6 |
| GDRO | 89.9 |
| AFR | 90.4 |
| SPARE | 89.8 |
| MaskTune | 86.4 |
| CIM | 77.2 |
| DFR | 91.8* |

Table 2: **Object-Centric Learning for Spurious Background OOD Generalization**. We report accuracy for each benchmark. "ImgNet-D (BG)" = ImageNet-D "background" subset; "ImgNet-9 (MR)" = ImageNet-9 "mixed rand" subset; "WGA" = worst group accuracy. O-H/O-D = OCCAM with HQES/FT-Dinosaur masks generator. ★ = state-of-the-art. See Table 5 for method citations.

leverages the disentangled representations for distinct objects: robust classification under spurious background cues. This subsection demonstrates that object masks are a simple but effective strategy to mitigate the influence of spurious correlations with backgrounds in classification tasks (Table 2).

**Setup.** We first describe our experimental setup, including datasets, metrics, and compared baselines. We use several standard datasets with spurious backgrounds or co-occurring objects — UrbanCars (Li et al., 2023), ImageNet-D (background subset) (Zhang et al., 2024), ImageNet-9 (mixed rand subset) (Xiao et al., 2021), Waterbirds (Sagawa et al., 2020), and CounterAnimals (Wang et al., 2024) — detailed further in §D. We measure model performance using the standard metric used in the respective benchmark: accuracy and worst group accuracy (WGA). We provide per-benchmark comparisons for reference, including results from other relevant methods, citing them alongside their names in the tables. We use the foundational segmentation model HQES (Lu et al., 2023) (O-H) and the state-of-the-art OCL method FT-Dinosaur (Didolkar et al., 2025) (O-D) for mask prediction in our training-free probe, OCCAM. We categorize methods with comparable image encoder backbones for fairness.

**Results.** Using masks significantly improves performance across all datasets, sometimes reaching 100% accuracy (e.g., on UrbanCars; Table 2(b)) or close to that performance on Waterbirds and ImageNet-9 (mixed rand) subsets. This shows the potential of simple, training-free object-centric

| Name | Mask Method | Mask Model | FG Detector | WB↑ | IN-9↑ | IN-D↑ | UC↑ | Cmn-Ctr↓ |
|------|-------------|------------|-------------|-----|-------|-------|-----|----------|
| CLIP | - | - | - | 83.6 | 91.9 | 17.6 | 87.2 | 15.0 |
|  | Gray BG + Crop | FT-Dinosaur | Ens. $\mathcal{H}$ | 83.8 | 84.0 | 52.4 | 95.2 | 13.1 |
|  |  |  | Class-Aided | 92.1 | 93.8 | 57.7 | 98.4 | 12.7 |
|  |  | HQES | Ens. $\mathcal{H}$ | 86.8 | 88.6 | 60.4 | 95.2 | 8.8 |
|  |  |  | Class-Aided | 96.0 | 95.2 | 68.0 | 100.0 | 8.5 |
| AlphaCLIP | - ($\alpha = 1$) | - | - | 79.8 | 90.2 | 23.5 | 87.2 | 17.0 |
|  | $\alpha$-channel | FT-Dinosaur | Ens. $\mathcal{H}$ | 81.0 | 90.3 | 40.7 | 92.0 | 17.2 |
|  |  |  | Class-Aided | 86.9 | 93.1 | 49.1 | 96.0 | 15.3 |
|  |  | HQES | Ens. $\mathcal{H}$ | 84.7 | 91.2 | 44.7 | 91.2 | 16.4 |
|  |  |  | Class-Aided | 89.1 | 93.1 | 53.9 | 97.6 | 15.2 |

Table 4: **Factor Analysis for Spurious Background OOD Generalization**. Accuracies on spurious correlations datasets when varying factors for the ViT-L-14 CLIP architecture. We use AlphaCLIP for $\alpha$-channel masking and CLIP for Gray Crop masking. We first report their baseline performances without masking (where mask method and model are both "-") and with 2 different mask models (FT-Dinosaur and HQES) as well as 2 different foreground detectors (Ens. $\mathcal{H}$ and Class-Aided). Results are reported on 5 benchmark datasets, Waterbirds (WB), ImageNet-9 (IN-9), ImageNet-D (IN-D), UrbanCars (UC), and CounterAnimals (Cmn-Ctr). For the CounterAnimals results, we report the gap between the common-split (Cmn) and the counter-split (Ctr) accuracies. Unlike other metrics, a smaller Cmn-Ctr gap is deemed a better generalization.

methods like OCCAM to address otherwise challenging downstream problems, if we can robustly identify the foreground object of interest. On harder benchmarks like ImageNet-D (background subset), HQES-based masks with SigLip models yield far better performance (78.5%) even compared to recent models like LLAVA 1.5 (Liu et al., 2023a) (73.3%), and outperform their best slot-based counterparts (71.5%) using FT-Dinosaur (Table 2(a)). Throughout, HQES consistently provides more effective masks than FT-Dinosaur.

**Conclusion.** These experiments show that mask-based, training-free object-centric probes can provide practical value on challenging robust classification tasks, if the task of foreground detection is sufficiently addressed (§3.2). It provides substantial gains on all tested benchmarks over the state-of-the-art methods for tackling spurious correlations. We hope this encourages the community to develop segmentation-based OCL approaches and demonstrate practical benefits across a variety of downstream applications. We next perform data-centric analysis leveraging properties of our OCL pipeline.

### 4.2.1 COUNTERANIMALS: SPURIOUS OR SIMPLY HARD?

Our object-centric classification pipeline can isolate an object's influence apart from its background. This property of OCL can be used to analyze the recently proposed CounterAnimals dataset (Wang et al., 2024).

**Setup.** CounterAnimals highlights models' reliance on spurious backgrounds. It consists of two splits from iNaturalist,[1] each containing animals from 45 classes in ImageNet-1k (Russakovsky et al., 2015). The Common split features typical backgrounds (e.g., polar bears on snow), while the Counter split features less common ones (e.g., polar bears on dirt). It primarily demonstrates that models consistently perform better on the Common than on Counter, due to spurious background cues.

| | CounterAnimals | |
|--------|----------------|----------------|
| Method | Cmn/Ctr (↑) | Cmn–Ctr (↓) |
| | AlphaCLIP ViT-L | |
| CLIP | 79.0/62.0 | 17.0 |
| O-D (Ours) | **85.8/70.5** | 15.3 |
| O-H (Ours) | 84.4/69.2 | 15.2 |

Table 3: **Data-Centric Understanding using OCL.** We report the accuracies on the Common and Counter subset of the CounterAnimals dataset. We see that after eliminating the spurious background using OCL methods, the gap (Cmn-Ctr) does not substantially decrease.

---

[1]https://www.inaturalist.org/observations

**What is the Contribution of Spurious Correlations?** We perform a simple check using OCCAM – If the drop from Common to Counter is caused by spurious background correlations, then using OCCAM we can ablate the contribution of everything except the foreground object. Ideally, ablating the background should result in roughly equal performance on both Common and Counter sets (the gap should be 0%). However, we see from Table 3, Table 4 and Figure 5 that even after ablating the background entirely, there is a substantial gap between the Common and Counter subsets. For example, when using AlphaCLIP, the gap reduces from 17.0% to 15.2%. Similarly, using HQES masks and a gray background for both sets, we still observe an 8.5% gap. This provides interesting evidence that images in the Common subset might be substantially easier than images from the Counter subset by about 8-10%.

**Conclusion.** OCL methods allow analyzing datasets, and analyse the contribution of individual objects. In the case of CounterAnimals, we find that spurious backgrounds might not be the primary reason the Counter subset is harder, although they are a factor. A significant (10%) gap might be caused by the Counter subset simply being harder to classify than the Common subset due to a wide variety of other factors. Overall, we show the potential for OCL methods to help inform data-centric fields like data attribution.

### 4.2.2 ABLATIONS: IDENTIFYING BOTTLENECKS IN OCCAM

We now ablate the contributions of different components in the OCCAM pipeline. We first test two CLIP models (CLIP and AlphaCLIP), to see whether our results generalize beyond simply removing backgrounds to recent techniques such as AlphaCLIP, which use the $\alpha$-channel to focus on the mask instead of eliminating the background. Secondly, we study the effect of the masking generator, testing HQES along with the current SOTA OCL method, FT-Dinosaur. Lastly, we study the influence of different FG Detection methods. We showcase our analysis in Table 4.

**Effect of mask applying method.** Using masks with Class-Aided FG detector improves performance on all the datasets for both Gray BG + Crop and $\alpha$-channel mask methods, but for the former, accuracy is usually higher. For example, on Waterbirds (Table 4), accuracy for the Gray BG + Crop mask method and the HQES mask generator is 96.0% while for AlphaCLIP it is 89.1%. This indicates that the backgrounds have strong spurious correlations that still affect $\alpha$-CLIP to a small extent.

**Effect of mask generator.** Comparing the rows from mask models to the original CLIP model, we see that both FT-Dinosaur and HQES improve performance, across CLIP and AlphaCLIP, given that we use Class-Aided FG detector. In this scenario, HQES improves accuracy more than FT-Dinosaur. For example, for the Gray BG + Crop mask method, it leads to 68.0% accuracy on ImageNet-D, while FT-Dinosaur reaches only 57.7%. This indicates that the segmentation-based OCL performs better consistently for downstream OCL applications.

**Selecting foreground mask.** Accuracy gains with Ens. $\mathcal{H}$ are always smaller than for Class-Aided FG detector and sometimes can be negative (Table 4). For example, for Gray BG + Crop mask method and HQES mask generator accuracy on ImageNet-9 drops from 91.9% to 88.6% when using Ens. $\mathcal{H}$ FG detector, while jumping to 95.2% with Class-Aided FG detector. Such results are not surprising at all, given that HQES with Class-Aided foreground detector is a very close approximation to classifying ground truth foreground objects (see § E for details). At the same time, this reveals a weakness in other baseline foreground detection methods and leaves room for improvement and future research.

**Conclusion.** The empirical results show that segmentation models outperform current OCL methods in obtaining object-centric representations that result in better classification. The simple Gray BG + Crop mask method generally performs better than the more advanced $\alpha$-channel mask method. At the same time, identifying foreground masks among many candidates remains a challenge.

## 5 DISCUSSION

**In defense of current OCL benchmarks.** One important aspect to clarify is the rationale behind the OCL researchers' choice to evaluate their models using object discovery benchmarks, as this may

not have been clearly articulated. Conventionally, OCL works have relied on constructing synthetic scenarios, where one has knowledge of the ground truth object-centric latent variables, e.g., object position, object color, etc, and can thus directly evaluate whether the learned representation encodes each object separately in its representation (Brady et al., 2023; Kori et al., 2023; Brady et al., 2024). One core aspect is scaling it to real-world scenarios, where we do not have knowledge of the data-generating process. Hence, traditional literature resorts to (a) probing the representation for object properties, such as object position, object color, etc (Arefin et al., 2024; Liu et al., 2023b), and (b) decoding slot representations to observe if they do indeed only possess a given object (Locatello et al., 2020; Jiang et al., 2023; Seitzer et al., 2023; Didolkar et al., 2025; Kipf et al., 2022; Elsayed et al., 2022; Greff et al., 2019).

**Should OCL be strictly unsupervised?** Traditionally, it was assumed that without access to auxiliary information or data-generating processes, there could be no ground-truth supervision for object-centric representations. Consequently, unsupervised learning — requiring no labels — became the standard approach for OCL. However, the advent of robust foundation models — that can leverage segmentation masks or text alongside images and generalize zero-shot across a wide range of inputs — now challenges the need for strict unsupervised constraints (Mamaghan et al., 2025). We believe OCL can greatly benefit from using all available data.

**Why not incorporate developmentally plausible multimodal cues in OCL?** When modeling human-like object perception, we should focus on developmentally plausible supervision. However, we note that the assumption of visual learning in infants being unsupervised also warrants reconsideration. Infants do not learn solely from static images; rather, they integrate a wealth of sensory cues (see Ayzenberg & Behrmann (2024) for a detailed review). For example, Spelke's seminal review (Spelke, 1990b) highlights the importance of dynamic information, such as motion and depth cues, for effective object segmentation in early development. Some object-centric works (e.g. Didolkar et al. (2023)) argue against this primarily based on the feasibility, citing the unavailability of multimodal data. However, there are several computational studies with models incorporating motion or depth (e.g. Karazija et al. (2022); Elsayed et al. (2022)), which also demonstrate that these additional cues can, in fact, be leveraged effectively. Thus, there is no inherent reason to confine OCL to strictly unsupervised, image-only paradigms when richer, multimodal data is often accessible in practice.

## 6 CONCLUSION AND OPEN PROBLEMS

Object-centric learning (OCL) is motivated by goals such as OOD generalization, sample-efficient composition, and cognitive insights. Yet progress is still measured mainly by object-discovery benchmarks. With strong segmentation methods like High-Quality Entity Segmentation (HQES) (Lu et al., 2023), class-agnostic models now surpass slot-based OCL in isolating objects, effectively achieving OCL's original aim.

However, its relevance extends beyond object discovery. We advocate for shifting OCL evaluation towards more realistic downstream tasks that leverage object-centric representations, such as mitigating spurious background correlations. We design a simple training-free probe, OCCAM, to show the efficacy of object-centric approaches to help classifiers generalise even in the presence of spurious correlations (§4.2), achieving near-perfect accuracies across many benchmarks (Table 2). By separating object-wise representation (well-addressed by HQES) from object selection (still a key challenge), OCCAM sheds light on where further improvements are needed.

We envision new OCL benchmarks advancing visual understanding via scene graphs, interpretable representations, and human-in-the-loop feedback. Diverse applications and benchmarks can drive progress, while OCL may also inform cognitive questions on how objects and causal structures emerge without supervision (Spelke, 1990a; Téglás et al., 2011).

## DISCLAIMER FOR USE OF LLMS

LLMs supported coding (experimentation, plotting) and writing refinement. The final version was carefully reviewed and finalized by the authors. LLMs were not used for ideation or experimental design.

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

## A  EXTENDED RELATED WORK

We extend § 2 with a detailed review of object-centric learning (OCL) from three perspectives: motivation, evaluation, and methodologies.

**Motivation for OCL.** The OCL community has inspired research from different perspectives. From one perspective, learning object-centric representations can help discover latent variables of the data-generating process, such as object position and color (Fumero et al., 2023), or even identify its causal mechanisms (Liu et al., 2023b; Schölkopf et al., 2021) by encoding structural knowledge that allows interventions and changes. From another perspective, OCL aims to simulate human cognition (Spelke, 1990a; Téglás et al., 2011; Wagemans, 2015) in neural networks. For example, infants intuitively understand physics by tracking objects with consistent behavior over time (Dittadi et al., 2022). They later reuse this knowledge to learn new tasks quickly. Advances in OCL can help neural networks develop this ability as well. In addition to that, some studies focus on understanding the compositional nature of scenes (Greff et al., 2020) by providing separate representations for different elements (e.g., human, hat, bed, table) and their interactions (a cat wearing a hat or a bear guiding cubs). Several papers claim that there is a potential to improve sample efficiency (Kapl et al., 2025) and generalization (Locatello et al., 2020; Kipf et al., 2022; Seitzer et al., 2023; Wiedemer et al., 2024; Mamaghan et al., 2025; Kapl et al., 2025) or object-centric methods can be more robust (Seitzer et al., 2023; Arefin et al., 2024). Others refer to the structure of the world, saying that the fundamental structure of the physical world is compositional and modular (Jiang et al., 2023) or that humans understand the world in terms of separate objects (Kipf et al., 2022; Didolkar et al., 2025). However, we have observed a consistent lack of empirical evidence demonstrating that object-centric approaches improve sample efficiency or aid in identifying causal mechanisms. To address this gap, we believe more empirical research is needed. As a first step, we show that robust classification is achievable even in the presence of explicitly distracting backgrounds and other object interference.

**OCL evaluation.** Measuring progress on the primary motivations of object-centric learning is a hard problem and suffers from a chronic lack of scalable benchmarks. Hence, empirical support for the commonly claimed benefits, such as parameter/learning efficiency (Kipf et al., 2022; Kapl et al., 2025) and improved generalization (Dittadi et al., 2022; Arefin et al., 2024; Wiedemer et al., 2024; Mamaghan et al., 2025; Kapl et al., 2025) or better understanding of representations, remains limited. Some papers study the link between object-centric learning and downstream applications. These include reinforcement learning (Watters et al., 2019; Kulkarni et al., 2019; Berner et al., 2019; Sun et al., 2019; Yoon et al., 2023), scene representation and generation (Kulkarni et al., 2019; El-Nouby et al., 2018; Matsumori et al., 2021; Burgess et al., 2019), reasoning (Webb et al., 2023; Yang et al., 2020), and planning (Migimatsu & Bohg, 2020). We highlight that these papers provide a valuable contribution to benchmarking progress in the OCL field. However, most research does not focus on these tasks. Much of the progress is tracked by unsupervised object discovery benchmarks, essentially entity segmentation (Locatello et al., 2020; Jiang et al., 2023; Seitzer et al., 2023; Didolkar et al., 2025; Kipf et al., 2022; Elsayed et al., 2022; Greff et al., 2019). Model performance is usually quantified with foreground adjusted random index (FG-ARI) (Rand, 1971; Hubert & Arabie, 1985; Kipf et al., 2022), which is a permutation-invariant clustering metric or mean best overlap (mBO) (Pont-Tuset et al., 2015; Seitzer et al., 2023). These evaluations primarily assess whether slots reliably isolate individual objects — a criterion we argue is overly restrictive in the broader context of object-centric learning. In our paper, we urge more work to additionally evaluate downstream applications, particularly given the emergence of foundational segmentation models that significantly outperform object-centric methods on standard object discovery tasks (see Table 1 and Figure 3).

**OCL methodologies.** OCL captured widespread attention with the introduction of SlotAttention (Locatello et al., 2020), which enabled iterative learning of separate latent representations for each object in an image. These latent "slots" can then be decoded back to the pixel space. Extensions have included SlotAttention paired with diffusion decoders (Jiang et al., 2023) and SlotAttention architectures built on top of DINO (Seitzer et al., 2023; Didolkar et al., 2025) features. Dinosaur (Seitzer et al., 2023) uses pre-trained self-supervised DINO (Caron et al., 2021) features as a target for reconstruction loss. This loss is used to train a decoder with Slot Attention (Locatello et al., 2020) on top of the ResNet (He et al., 2016) encoder. FT-Dinosaur (Didolkar et al., 2025) improves Dinosaur by replacing the ResNet encoder with a DINO-ViT (Dosovitskiy et al., 2021) encoder sep-

arate from the one used to compute target features. It jointly fine-tunes the encoder with the decoder. SlotDiffusion (Jiang et al., 2023) uses pre-trained features from the Stable Diffusion Encoder (Rombach et al., 2022) and trains a diffusion-based decoder with Slot Attention (Locatello et al., 2020) on top of them. In video contexts, sequential adaptations leverage temporal dependencies (Kipf et al., 2022) and depth information (Elsayed et al., 2022). Some studies also propose theoretical foundations for OCL (Wiedemer et al., 2024; Brady et al., 2023). There is also a line of work that studies object-centric representation in the context of out-of-distribution (OOD) generalization in segmentation (Dittadi et al., 2022), compositional generalization (Wiedemer et al., 2024; Mamaghan et al., 2025; Kapl et al., 2025), and classification, e.g., CoBalT (Arefin et al., 2024) that employs model distillation and slots clustering into concepts to refine feature quality. In our experiments, we compare with the latest methods – SlotDiffusion (Jiang et al., 2023) and (FT-)Dinosaur (Seitzer et al., 2023; Didolkar et al., 2025) for object discovery and CoBalT (Arefin et al., 2024) across robust classification benchmarks.

| Method | Citation |
| --- | --- |
| ImgNet-D (BG) | (Zhang et al., 2024) |
| ImgNet-9 (MR) | (Xiao et al., 2021) |
| UrbanCars | (Li et al., 2023) |
| Waterbirds | (Sagawa et al., 2020) |
| CLIP | (Radford et al., 2021) |
| CLIP-SigLip | (Zhai et al., 2023) |
| MiniGPT-4 | (Zhu et al., 2024) |
| LLaVa | (Liu et al., 2023a) |
| LLaVa-NeXT | (Liu et al., 2024b) |
| LLaVa-1.5 | (Liu et al., 2024a) |
| CoBalT | (Arefin et al., 2024) |
| LfF | (Nam et al., 2020) |
| JTT | (Liu et al., 2021) |
| SPARE | (Yang et al., 2023) |
| LLE | (Li et al., 2023) |
| SIN / INSIN / INCGN | (Sauer & Geiger, 2021) |
| MaskTune | (Asgari et al., 2022) |
| CIM | (Taghanaki et al., 2021) |
| GDRO | (Sagawa et al., 2020) |
| AFR | (Qiu et al., 2023) |
| DFR | (Kirichenko et al., 2023) |

Table 5: Mapping from methods to their original citations.

## B    FOREGROUND DETECTORS COMPARISON

To justify the choice of $g_{\text{class\_aided}}$ and $g_{\mathcal{H}}$ in § 3.2, we compare several foreground detection methods. One can notice that foreground detection is an application of an out-of-distribution (OOD) detection, a well-studied problem (Mukhoti et al., 2021; Tran et al., 2022; Gruber & Buettner, 2022) — with foreground objects treated as in-distribution (ID) samples and background objects as OOD samples. Hence, we evaluate OOD detection methods for this task in Figure 4.

**Setup.** We construct an OOD detection dataset using the ImageNet-1k (Russakovsky et al., 2015) validation set by leveraging ground truth bounding boxes[2] to derive accurate foreground masks (see details in § G). Performance is measured via the area under the ROC curve (AUROC), in line with standard OOD detection frameworks (Mukhoti et al., 2021; Tran et al., 2022; Gruber & Buettner, 2022; Mucsányi et al., 2024; Rubinstein et al., 2024). We use the following strong baselines:

- *Class-Aided (single model)* (Hendrycks & Gimpel, 2017): $p^y(x)$
- *Ensemble entropy* (Ovadia et al., 2019): $\frac{1}{M} \sum_{k=1}^{M} \mathcal{H}[p_k(x)]$
- *Ensemble confidence* (Lakshminarayanan et al., 2017): $\max_c \frac{1}{M} \sum_{k=1}^{M} p_k^c(x)$
- *Confidence (single model)* (Hendrycks & Gimpel, 2017): $\max_c p^c(x)$
- *Entropy (single model)* (Depeweg et al., 2017): $\mathcal{H}[p(x)]$

Here, $y$ is ground truth label, $p(x)$ denotes the model's probability vector prediction for the corresponding sample $x$, $M$ is the ensemble size, and $\mathcal{H}$ represents entropy. We use the ViT-L-14 CLIP model pre-trained by OpenAI (Radford et al., 2021) as the single model, and 5 CLIP models with ViT-L-14 (Dosovitskiy et al., 2021) vision encoders pre-trained on different datasets as the ensemble. Note that OpenAI ViT-L-14 was the strongest model by AUROC among the ensemble, hence it was used as the single model. Further details are provided in § G.

**Results.** As shown in Figure 4, Class-Aided achieves the highest AUROC of $90.1\%$ whereas the ensemble entropy method yields $89.6\%$. Other methods perform significantly worse. Nevertheless, all methods score more than $80\%$ AUROC.

**Conclusion.** The AUROC performance of Class-Aided and Ens. $\mathcal{H}$ foreground detectors showed only minor differences from each other, both scoring around $90\%$ and being the best among the compared methods; however, substantial performance gaps remain when comparing the Class-Aided results with the Ens. $\mathcal{H}$ foreground detector in spurious correlation tasks (Table 4), a possible reason for this is discussed in § E. This disparity highlights two key implications. Current evaluation metrics may have a large research gap to better reflect real-world applications.

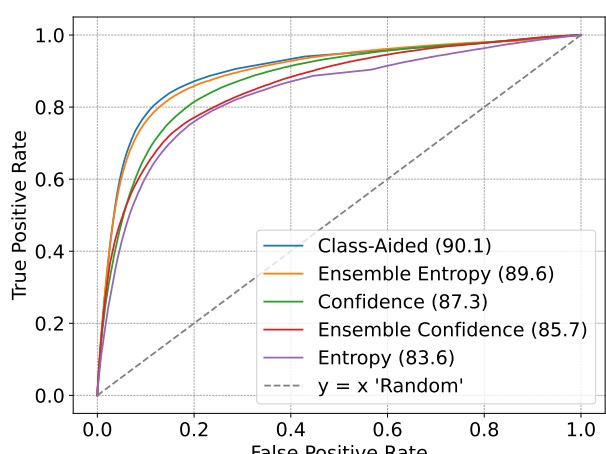

Figure 4: **Foreground Object Detection.** ROC-curves for foreground detection methods. For each scoring scheme, we measure how well the true foreground objects in the ImageNet-validation dataset are detected. More details in § G.

Conversely, spurious correlation foreground detection might be a promising proxy task for identifying better OOD detection models.

[2]https://academictorrents.com/details/dfa9ab25

## C COUNTERANIMALS: GAPS BETWEEN "COMMON" AND "COUNTER" SUBSETS

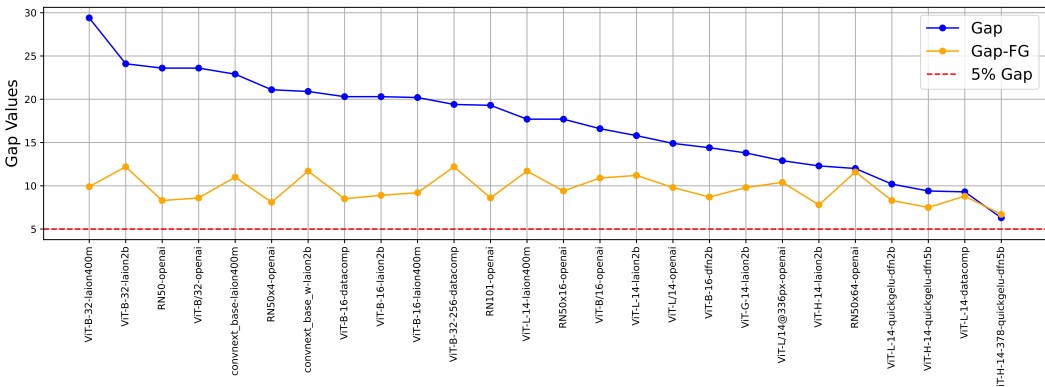

Figure 5: Gaps in accuracies [Common - Counter] for Common and Counter subsets of CounterAnimals (Wang et al., 2024) dataset correspondingly for different CLIP models and pre-training datasets. "Gap" results are computed for CLIP (Radford et al., 2021) zero-shot performance without using any masks; "Gap-FG" results are computed when using OCCAM with HQES (Lu et al., 2023) masks, Class-Aided foreground selection method, and "Gray BG + Crop" mask applying operation.

In addition to the results in Table 2 (e), we present the complete performance results for all CLIP models from the original CounterAnimals dataset (Wang et al., 2024) in Figure 5. This figure illustrates the performance gaps between the Common and Counter subsets, as discussed in § 4.2.1.

We observe that the performance gaps are consistently greater than 5%, as all points lie above the red dashed line. For some models, such as ViT-L-14-datacomp and ViT-H-14-quickgelu-dfn5b, the gaps remain nearly unchanged with or without using OCCAM — around 10% and 6%, respectively.

We argue that for these models, the original gaps reported in the CounterAnimals paper (Wang et al., 2024) (referred to as "Gap" in our notation) cannot be attributed solely to the models' reliance on spurious background cues. This is because the gap remains even after background removal using the "Gray BG + crop" masking operation (referred to as "Gap-FG" in our notation).

## D  DETAILS ON SPURIOUS BACKGROUNDS DATASETS

Below we provide details on the datasets used in our study (for more information on the CounterAnimals dataset, see § C):

The core of our dataset collection includes several widely-used benchmarks for evaluating robust image classification models: UrbanCars (Li et al., 2023), Waterbirds (Sagawa et al., 2020), and ImageNet-9 (Xiao et al., 2021). We also include the ImageNet-D dataset (Zhang et al., 2024), which we consider to offer more realistic visual compositions, as it uses a diffusion model (Rombach et al., 2022) to blend objects with backgrounds, rather than relying on manual cut-and-paste techniques as in the previous datasets. Finally, we use the CounterAnimals dataset (Wang et al., 2024), a recently introduced benchmark consisting of natural images with spurious background correlations, specifically designed to challenge even CLIP models.

1. **UrbanCars** (Li et al., 2023): A binary classification dataset that categorizes cars as either "urban" or "country." Each image contains a car paired with a contextually related secondary object (e.g., a fire hydrant for urban or a cow for country) and is placed on either an urban or rural background. All elements are synthetically combined from cut-out components.

2. **ImageNet-D** (Zhang et al., 2024): A synthetic dataset generated using diffusion models for 113 ImageNet-based classes (a subset of ImageNet-1k (Russakovsky et al., 2015)). We focus on the "background" subset, where objects appear in unexpected contexts (e.g., plates in a swimming pool), to test robustness to spurious background cues.

3. **ImageNet-9** (Xiao et al., 2021): A synthetic dataset with 9 broad object categories (e.g., dog, bird), each corresponding to supersets of ImageNet classes. We use the "mixed random" subset, where objects are placed on backgrounds from different, unrelated classes.

4. **Waterbirds** (Sagawa et al., 2020): A binary classification dataset where bird species are labeled as either "land" or "sea" birds. Each image features a bird placed on either a land or sea background. Like UrbanCars, this dataset is synthetically constructed using cut-out birds and backgrounds.

# E CLASS-AIDED FOREGROUND DETECTOR YIELDS THE CLOSEST APPROXIMATION TO GROUND TRUTH FOREGROUND MASKS

| FG Detector | WGA ($\uparrow$) |
|---|---|
| - | 83.6 |
| Max Prob | 78.6 |
| Ens. $\mathcal{H}$ | 86.8 |
| Class-Aided | 96.0 |
| Ground Truth | **96.7** |

Table 6: **Different foreground detectors on Waterbirds** We report the worst-group accuracies on the Waterbirds dataset for different foreground detectors. Masks are generated by HQES and applied via "Gray BG + Crop" (see § 3.1). The classification model is CLIP ViT-L-14 (Radford et al., 2021). "-" stands for classification of original images without using any masks. Max Prob stands for foreground detector that uses the following score (in terms of § 3.2): $g_{\text{max\_prob}}(x, m) = \max_c p^c(\psi(a(x, m)))$ - maximum probability across all possible classes (its computation is equivalent to confidence in § B). Class-Aided and Ens. $\mathcal{H}$ are described in § 3.2. Ground Truth stands for ground truth foreground masks that are taken from (Kirichenko et al., 2023).

The Class-Aided foreground detector selects masks based on the highest ground truth class probability (§ 3.2).

Such a strategy may introduce a selection bias towards non-foreground masks that boost the overall classification accuracy of OCCAM, but are unrelated to the actual objects of interest — for example, masks highlighting spurious background regions that correlate with the ground truth label. For this reason, we were initially cautious about treating it as a reliable foreground detector.

However, on the Waterbirds dataset (Sagawa et al., 2020), for which ground truth foreground masks are available (Kirichenko et al., 2023), we find that this bias is infrequent. In a random sample of 100 images, Class-Aided selected a non-foreground mask in only 5 cases. Despite this, the classification accuracy using Class-Aided masks is 96.0%, only slightly lower than the 96.7% achieved with ground truth masks (see Table 6).

Based on this, we do not observe strong evidence that the Class-Aided detector frequently selects non-foreground masks, whereas we find that the selected masks perform comparably to ground truth in the context of classification under spurious correlations. Therefore, we consider the masks chosen by the Class-Aided foreground detector to be the closest available approximation to ground truth foreground masks in the absence of mask supervision.

# F EXTENDED IMPLEMENTATION DETAILS

**Classes for zero-shot classification** Following the original CLIP (Radford et al., 2021) work, we compute the classifier's pre-softmax logits $f(\psi(x))$ using dot products between image embeddings and text embeddings of class name prompts. Each prompt follows the format: "A photo of $X$", where $X$ is a class name from the corresponding dataset.

For Waterbirds (Sagawa et al., 2020) and UrbanCars (Li et al., 2023), we first compute dot products using prompts based on fine-grained class names from the Caltech Birds (CUB) dataset (Wah et al., 2011) and the Stanford Cars dataset (Krause et al., 2013), respectively. This is because the foreground objects in these datasets were originally cropped from the corresponding source datasets.

All fine-grained classes are then grouped into two broader categories. For Waterbirds, the classes are divided into "land" and "sea" birds. For UrbanCars, they are grouped into "urban" and "country" cars. The final prediction corresponds to the group containing the fine-grained class with the highest dot product.

**How resize is done for "Gray BG + Crop"** We apply the following steps to perform the "Gray BG + Crop" operation: (1) Find the smallest rectangle that fully contains the foreground object. (2) Expand the shorter side of this rectangle to match the longer side, ensuring that the center of the new square matches the original rectangle's center. (3) Resize the resulting square to the target resolution.

**Fixed number of slots in OCL method** When using FT-Dinosaur (Didolkar et al., 2025) as a mask generator in the OCL method, we fix the number of slots to 5, following the recommendation from the original implementation.

**Foundational segmentation model choice** While HQES and SAM generally perform similarly on segmentation tasks, SAM shows significantly better performance on the mBO metric. Despite this, we use HQES in all of our main experiments, as we have full knowledge of its training data and can confirm that it was not trained on any of the datasets containing spurious correlations used in our evaluation.

**Mask-free AlphaCLIP** AlphaCLIP (Sun et al., 2023) requires a foreground mask as input. To simulate a mask-free setting, we use a mask that covers the entire image, effectively setting $\alpha = 1$. Although a mask is technically provided, it does not contain any useful localization information, so we treat this setup as mask-free performance.

**Masks filtering** Before using masks in our experiments, we apply the following filtering rules:

1. **Size:** Remove masks that cover less than $0.001$ of the image pixels.
2. **Connected components:** Remove masks that contain more than $30$ connected components.
3. **Background heuristic:** Remove masks that cover at least $6$ of the $8$ key points (the $4$ corners and the $4$ side centers of the image).

# G    ADDITIONAL DETAILS ON FG DETECTORS COMPARISON

In this section, we give additional details on comparing different candidates for FG detector methods apart from $g_{\text{class\_aided}}$ and $g_{\mathcal{H}}$ (see § 3.2 for details).

**Dataset construction details.** We construct a binary classification dataset using the ImageNet validation set (Russakovsky et al., 2015), considering only images that have ground truth bounding boxes for the main object (i.e., the one corresponding to the ground truth label). For each such image, we predict masks for all objects it contains, as described in the "Generating masks" paragraph in §3.1. We then apply each mask using the "Gray BG + Crop" operation, following the "Applying masks" paragraph in §3.1. Each resulting masked image is assigned a label as follows:

- Class 1 (foreground) if its corresponding mask has the highest Intersection over Union (IoU; (Rezatofighi et al., 2019)) with the ground truth bounding box.
- Class 0 (non-foreground) otherwise.

**How are OOD detectors used?** OOD detectors are used in the following way: First, we compute an uncertainty score for each sample using formulas from § B based on the ensemble's outputs (for single model entropy and Ens. $\mathcal{H}$ we additionally multiply this score by $-1$ so that it is lower for OOD samples than for ID samples). Then, we treat this uncertainty score as the probability of predicting class 1 in our binary classification setting.

Note: Ens. $\mathcal{H}$ corresponds to $g_{\mathcal{H}}$ and Class-Aided corresponds to $g_{\text{class\_aided}}$, as described in the "Foreground detector" paragraph in § 3.2.

**Ensemble members.** All model checkpoints are sourced from the "openclip" library (Ilharco et al., 2021), using the following pre-training dataset identifiers: "openai", "datacomp_xl_s13b_b90k", "dfn2b", "laion400m_e31", and "laion400m_e32".

We focus on ensemble-based baselines for OOD detection, as they are among the most competitive approaches for this task (Mukhoti et al., 2021; Ovadia et al., 2019).

