# OpenReview forum: "From Slots to Masks: Rethinking OCL"
_ICLR.cc/2026/Conference — ICLR 2026 Conference Withdrawn Submission_

### Official Review · Reviewer_H62M · 2025-10-29

**Soundness:** 2
**Presentation:** 2
**Contribution:** 2
**Rating:** 2
**Confidence:** 4

**Summary:**

This manuscript aims to evaluate object-centric learning methods beyond conventional mask-based metrics, examining object-centric representations and their downstream utility. Object-centric representations are extracted from pre-trained vision encoder outputs through masking and pooling, then assess them on classification tasks such as UrbanCars using linear probing. The manuscript also argues that with the availability of foundation models like Segment-Anything (SAM), the current emphasis on mask supervision restrictions in object-centric learning may be less justified.

**Strengths:**

Evaluating object-centric learning methods through representation quality rather than solely mask quality is a promising and practically relevant direction. The idea of connecting object-centric representations with downstream task performance aligns with real-world utility. In addition, the discussion about leveraging foundation models for object-centric segmentation is timely and well-motivated.

**Weaknesses:**

The manuscript lacks a comprehensive literature review. Important related works, such as CutLER [1], UnSAM [2], and U2Seg [3], which perform unsupervised object segmentation and evaluate their masks using COCO mAP under both class-agnostic and class-specific settings, are not discussed. These works already offer more practical and challenging evaluation protocols. Given these works, the manuscript provides limited new insights beyond restating the potential of foundation models like SAM. Also, the choice of SAM is questionable under an unsupervised learning context, as unSAM would provide a fairer comparison. Overall, the manuscript reads more as a workshop submission rather than one in the main track.




References:

[1] Wang et al. Cut and Learn for Unsupervised Object Detection and Instance Segmentation (CutLER). CVPR. 2023.

[2] Wang et al. Segment Anything without Supervision (UnSAM). NeurIPS. 2024.

[3] Niu et al. Unsupervised Universal Image Segmentation (U2Seg). CVPR. 2024.

**Questions:**

1. Why was SAM chosen instead of unSAM, which is more consistent with unsupervised object-centric learning?

2. How does the proposed evaluation framework offer new insights beyond existing works (e.g., CutLER and u2Seg) that already benchmark unsupervised object masks?

---

### Official Review · Reviewer_LQCp · 2025-10-31

**Soundness:** 2
**Presentation:** 2
**Contribution:** 2
**Rating:** 2
**Confidence:** 3

**Summary:**

This paper revisits the paradigm of Object-Centric Learning (OCL) and argues that object separation—long considered the central challenge of OCL—has been effectively solved by modern class-agnostic segmentation models such as HQES and SAM. The authors propose a simple, training-free probe called OCCAM (Object-Centric Classification with Applied Masks) to test whether segmentation-based representations can improve out-of-distribution (OOD) robustness in classification tasks with spurious background correlations. Extensive experiments show strong zero-shot performance and suggest that segmentation-based methods outperform traditional slot-based OCL approaches.

**Strengths:**

1.The paper courageously questions the long-standing assumption that slot-based object separation remains an open challenge, suggesting that OCL should instead focus on downstream generalization and robustness.
2.The proposed OCCAM pipeline is simple, transparent, and easy to reproduce. The design choice of a training-free approach allows readers to directly see the empirical gap between classical OCL and segmentation-based solutions.
3.The experiments cover a wide range of OOD robustness benchmarks (e.g., Waterbirds, ImageNet-D, UrbanCars), and the results are consistent across architectures. The qualitative and quantitative comparisons are well presented.

**Weaknesses:**

1.Although the paper argues that segmentation models “solve” object discovery, the comparison is somewhat unbalanced. HQES and SAM are massive, fully supervised models trained on millions of labeled images, whereas the compared OCL baselines (e.g., SlotDiffusion, Dinosaur) are unsupervised and trained on small synthetic datasets. This asymmetry weakens the central claim that segmentation replaces slot-based OCL.
2.As acknowledged in the paper itself (§2), OCL was originally motivated by broader cognitive and causal reasoning goals—not merely object segmentation. Therefore, demonstrating that segmentation improves robustness on classification tasks does not necessarily invalidate or replace slot-based approaches aimed at compositional or causal learning. The work’s scope is narrower than its rhetorical ambition.
3.OCCAM’s design is rather straightforward: it works by masking an image and then classifying the cropped foreground. The “Gray BG + Crop” operation is effective, but it appears ad hoc and lacks theoretical justification for why it should emulate object-centric reasoning.
4.The use of “Class-Aided foreground scores” (which depend on ground truth labels) introduces bias and undermines claims of a training-free or unsupervised probe. In real-world zero-shot applications, this signal would not be available, so the method’s robustness in a purely unsupervised setting remains uncertain.
5.The paper occasionally implies that segmentation-based approaches “replace” OCL or “solve” object discovery entirely. Given that object compositionality, reasoning, and causal inference remain unresolved, such claims feel premature.

**Questions:**

1.How does the method perform on tasks beyond classification, such as reasoning, compositionality, or scene manipulation? These tasks are also other core motivations for OCL.
2.Could the observed improvements be attributed simply to stronger priors from HQES/SAM rather than genuine object-centric understanding?
3.The comparisons contrast fully supervised, large-scale segmentation models (e.g., HQES, SAM) with unsupervised OCL methods trained on synthetic data. How do the authors justify that such comparisons reflect a fair assessment of OCL progress rather than a demonstration of dataset and scale effects?
4.The “Gray BG + Crop” operation is manually designed and may bias results toward visual saliency rather than semantic relevance. Have the authors tested alternative masking strategies to validate the robustness of the claimed effects?

---

### Official Review · Reviewer_Uody · 2025-10-31

**Soundness:** 2
**Presentation:** 3
**Contribution:** 2
**Rating:** 2
**Confidence:** 3

**Summary:**

The paper proposes a segmentation-based feed-forward pipeline to obtain an object-centric representation without training. More specifically, a mask generator, such as SAM, is employed to generate object-wise masks. Next, the masks are used to mask out specific objects, which are consequently fed into an image encoder. A class-aided or ensemble-based foreground score is used to determine the final object-centric representation. The paper demonstrates that this segmentation-based approach outperforms object-centric learning, such as slot-based learning.

**Strengths:**

- The paper is well & clearly written, the method can be easily understood
- Training-Free approaches are generally valuable since their low computational cost makes them accessible
- The connection between segmentation and the output object-centric models is intuitive
- The overall presentation of the paper is good

**Weaknesses:**

- The technical contribution of the proposed method are rather slim. Instead, the approach pieces together off-the-shelf components to achieve the task
- The methodical connection to object-centric learning remains unclear. The approach is a simple inference pipeline, filtering images for objects and encoding them into a vector representation.
- The class-aided foreground score requires the ground-truth class to be known. In practice, as the authors correctly note, this is not feasible. Hence, this approach should not be part of the method. Instead, the approach should focus on the ensemble entropy.
- The connection between the text and Figure 1 remains unclear. The figure is not mentioned in the text nor is the topic of scene graphic in relation to the proposed approach much discussed

**Questions:**

- How does the topic of scene graphs relate to the proposed method? / Why is the Figure positioned as a Teaser figure given this lack of connection?
- In a practical setting, what could be a substitute for the ground truth class (which is not available in practice) to obtain a class-aided foreground score?
- Maybe I have missed it, but what specific image encoder is used to encode the masked object representation? What is the model trained with?

---

### Official Review · Reviewer_Tnbz · 2025-11-01

**Soundness:** 4
**Presentation:** 3
**Contribution:** 2
**Rating:** 6
**Confidence:** 4

**Summary:**

The paper’s motivating observation is that current general segmentation methods outperform specialized slot-centric object-centric learning methods. Based on this observation, it proposes a simple method to leverage general class-agnostic segmentation models for object-centric classification by masking detected objects, classifying the likely foreground object, and classifying the masked likely foreground object. Finally, the paper investigates robustness on out-of-distribution classification tasks.

**Strengths:**

The paper makes an interesting point in that existing slot-centric OCL methods perform worse than general-purpose class-agnostic segmentation models at unsupervised object discovery. The proposed method for leveraging class-agnostic segmentation models is maximally simple. Evaluating on OOD classification robustness tasks is highly interesting. I especially like the CounterAnimals evaluation, since it illustrates potential real-world impact, and a method to disentangle the impact of spurious correlations from dataset difficulty.

**Weaknesses:**

The proposed method to leverage segmentation models as described is extremely simple and not particularly novel, including the observation that this can lead to improved OOD generalization (e.g. [1]). Still, I do think the value of the work lies primarily in identifying the shortcomings in existing OCL literature and highlighting a path towards more meaningful evaluation setups.

[1] https://openaccess.thecvf.com/content/ICCV2023W/OODCV/papers/Aniraj_Masking_Strategies_for_Background_Bias_Removal_in_Computer_Vision_Models_ICCVW_2023_paper.pdf

**Questions:**

What was the reasoning behind using High-Quality Entity Segmentation (HQES) and FT-Dinosaur for the final experiments?

---

### Official Review · Reviewer_rADP · 2025-11-02

**Soundness:** 2
**Presentation:** 2
**Contribution:** 2
**Rating:** 4
**Confidence:** 5

**Summary:**

This paper presents a perspective on the current state of Object-Centric Learning (OCL), arguing that the field's preoccupation with unsupervised object discovery using slot-based mechanisms is becoming obsolete due to the advent of powerful, zero-shot segmentation models. The authors substantiate this claim by showing that models like HQES and SAM outperform traditional OCL methods on standard discovery benchmarks. Proposing a shift in focus towards downstream applications, the paper introduces OCCAM, a training-free pipeline that leverages segmentation masks to improve OOD robustness in classification tasks, particularly against spurious background correlations.

**Strengths:**

1. The paper argues that slot-based Object-Centric Learning methods are overly focused on the task of unsupervised object discovery, a problem that is being better solved by modern, off-the-shelf, class-agnostic segmentation models (e.g., SAM, HQES) with superior zero-shot performance.

2. The paper introduces a training-free probe named OCCAM. This pipeline first uses a segmentation model to generate object masks, then applies these masks to isolate objects, and finally classifies the isolated object to improve robustness against spurious background correlations.

3. On object discovery tasks, segmentation models like HQES significantly outperform state-of-the-art slot-based OCL methods.

**Weaknesses:**

1. The paper's motivation is to "rethink OCL" and call for a paradigm shift. However, its core technical contribution, OCCAM, is essentially a very simple, non-end-to-end pipeline stitched together from off-the-shelf components (a segmentation model + a classifier). This technical solution itself lacks depth and novelty and does not match the grand narrative of "Rethinking OCL." It feels more like a "probe" for validating an idea or a strong baseline, rather than a novel methodology.

2. The most impressive performance gains reported in the paper (e.g., O-H/O-D in Table 2) are mostly based on the Class-Aided foreground detector. This detector requires access to the ground-truth class label to select the optimal mask, which is an "oracle" condition that cannot be met in real-world applications. In contrast, the more realistic Ens. H detector performs inconsistently and, in some cases, even worse than the baseline (see Table 4, where accuracy on ImageNet-9 drops from 91.9% to 88.6%). This reveals a huge gap: the "potential" claimed by the paper is built on an unrealistic assumption, while its practical, usable method is not consistently effective. This severely undermines the persuasiveness of its conclusions.

3. Traditional slot-based OCL aims to learn disentangled representations within a unified, end-to-end, and often unsupervised framework. The "segmentation + classification" pipeline proposed here relies on large-scale supervised pre-trained models and is a multi-stage process. This isn't a "rethinking" or "improvement" of OCL. While this point has its merits, it sidesteps the core scientific question of the OCL field: how to learn structured representations of the world without strong supervision.

4. I believe that the author should transcend the limitation of segmented tasks and discuss more about the actual applications of OCL in the open world [1,3,5], while also engaging in comparative discussions with the current work [2,4].

[1] SELDOM: Scene Editing via Latent Diffusion with Object-centric Modifications. In ICCV, 2025.

[2] Dissecting Generalized Category Discovery: Multiplex Consensus under Self-Deconstruction. In ICCV, 2025.

[3] OCRT: Boosting Foundation Models in the Open World with Object-Concept-Relation Triad. In CVPR, 2025.

[4] SlotPi: Physics-informed Object-centric Reasoning Models. In KDD, 2025.

[5] Reconstruct and Match: Out-of-Distribution Robustness via Topological Homogeneity. In NeurIPS, 2025.

**Questions:**

1. Given the limited technical novelty of the OCCAM pipeline, how do the authors position the core contribution of this paper? Is it a position paper that points out a problem in the field, a comprehensive benchmark study, or the proposal of a novel method? If it is positioned as a novel method, please clarify its core algorithmic innovation that goes beyond a simple combination of existing tools.

2. Many of OCL's motivations, such as learning causal structures, compositional generalization, and achieving human-like cognition, are closely tied to end-to-end, emergent representations. To what extent can a modular pipeline that relies on external, strongly supervised tools still claim to be pursuing these original goals? Does this shift "from slots to masks" bring us closer to or further away from the ultimate goal of understanding and building an agent that can autonomously learn the structure of the world?

---

### Note · Authors · 2025-11-12

I have read and agree with the venue's withdrawal policy on behalf of myself and my co-authors.